# OpenReview forum: "What Makes Large Language Models Undistillable?"
_ICLR.cc/2026/Conference — ICLR 2026 Conference Withdrawn Submission_

### Official Review · Reviewer_FMXm · 2025-10-26

**Soundness:** 2
**Presentation:** 2
**Contribution:** 2
**Rating:** 2
**Confidence:** 3

**Summary:**

This paper investigates the phenomenon of Knowledge Distillation failure in LLMs, seeking to answer the question: "What makes an LLM undistillable?" The authors introduce the concept of a "distillation trap," where a high-performing teacher model generates outputs that are linguistically coherent but contain nonsensical or flawed reasoning, which misguides the student model during training. To study this, the paper proposes a novel method to actively engineer this failure mode, transforming standard LLMs into "undistillable teachers." The authors demonstrate empirically across multiple model pairs and reasoning datasets that this method causes a catastrophic performance collapse in student models trained via state-of-the-art KD, thereby validating their hypothesis.

**Strengths:**

1.  The primary strength is its "constructive proof" approach. Instead of merely observing KD failures and attempting to patch them, the authors actively engineer the failure mode. The framing of the "distillation trap" is clear, intuitive, and provides a valuable new lens through which to view the dynamics of knowledge transfer.

2.  The proposed RFT-based method is both clever and well-motivated. The use of a smaller, frozen reference model to define the "trap reward"  is a particularly elegant solution. It operationalizes the abstract concept of "confusing reasoning" into a concrete, computable metric without the need for human annotation or more complex heuristics. This design choice is both theoretically sound and computationally practical.

**Weaknesses:**

1.  The paper briefly mentions that the framework could be used to *enhance* distillability by using a negative weight for the trap reward (λ = -1) and provides a single qualitative example. This is a fascinating and potentially very impactful claim, but it is not supported by any quantitative evidence. Without experiments showing that a student distilled from such an "enhanced" teacher outperforms a student distilled from the original teacher, this remains a speculative assertion.

2. The core assumption behind the trap reward is that "correct and valuable reasoning should be self-concordant" and thus not surprising to a smaller model. While this is a reasonable and effective heuristic, it may not be universally true. A particularly novel, complex, or "out-of-the-box" correct reasoning path might also register as high-entropy to a smaller reference model. The current method risks inadvertently penalizing such creative-yet-correct solutions.

3. The paper also strongly assumes that the proposed distillation trap is the sole reason why LLM cannot be distilled, and then conducts experiments based on this "belief." I find this logical relationship confusing. The current approach simply demonstrates that the teacher model can first speak a potentially irrelevant sentence, then "adaptively" ignore this irrelevant/illusory content and directly output the correct answer. It's obvious that this data construction method will prevent the student model from learning. There are many reasons for distillation failure, but the paper does not provide any other relevant discussion.

**Questions:**

The authors attempt to demonstrate that this affects the effectiveness of knowledge distillation by constructing data that is "unlearnable." So, if we remove all such data from the existing training set, will LLM be able to perform better distillation? I think this experiment would be more straightforward.

---

### Official Review · Reviewer_mBKq · 2025-10-28

**Soundness:** 2
**Presentation:** 3
**Contribution:** 2
**Rating:** 4
**Confidence:** 4

**Summary:**

The paper explores when and why Knowledge Distillation (KD) fails for large language models (LLMs). The authors formalize a new failure mode called the “distillation trap,” in which a teacher model generates outputs that are superficially coherent yet semantically misleading, causing the student’s learning to diverge. They trace this to the optimization dynamics of KL divergence (forward vs reverse), showing how these divergences are vulnerable to deceptive modes. To validate their theory and to provide a tool, the authors propose a Reinforcement Fine-Tuning (RFT) method to make a teacher model undistillable. Experiments on multiple teacher-student pairs and datasets (gsm8k, CSQA, MMLU-Pro, superGPQA) show that these undistillable teachers preserve their original task performance (small deviations) while resulting in catastrophic performance drops on their corresponding distilled students.

**Strengths:**

- Novel perspective on distillation: The paper takes an original and thought-provoking approach by shifting the focus from improving student performance to protecting the teacher model’s knowledge, introducing the concept of “distillation traps” as a new paradigm in model robustness and intellectual property protection.

- Clarity and presentation: The paper is well written, clearly structured, and easy to follow, with solid motivation and intuition provided for each component of the proposed framework.

- Empirical effectiveness and generalization: The experimental results convincingly demonstrate that the proposed method successfully makes the teacher model undistillable while maintaining strong performance, and the approach generalizes well to out-of-distribution data.

**Weaknesses:**

- The authors focus solely on making the teacher undistillable but ignore the inverse direction, improving the teacher through boosted distillation. Including experiments to evaluate this setup could provide a more complete understanding of the proposed framework.

- While the experiments demonstrate the effectiveness of the approach, the results may be conditioned on the specific student model used during training. It would strengthen the paper to test whether the distillation trap generalizes to other student architectures or capacities.

 - The undistillable version of the teacher exhibits hallucination behavior, which is undesirable for generative models. This raises concerns about potential trade-offs between making a model undistillable and maintaining its generation quality.

**Questions:**

See weaknesses section

---

### Official Review · Reviewer_aA1C · 2025-10-31

**Soundness:** 2
**Presentation:** 3
**Contribution:** 2
**Rating:** 2
**Confidence:** 3

**Summary:**

This paper investigates why Knowledge Distillation (KD) sometimes fails for Large Language Models (LLMs). It identifies and formalizes the "distillation trap," where a high-performing teacher model generates outputs that are linguistically coherent but nonsensical or hallucinatory, thus misguiding the student model.

The authors link this trap to the dynamics of the Kullback-Leibler (KL) divergence loss function. They then propose a Reinforcement Fine-tuning (RFT) method to intentionally amplify these traps, creating "undistillable teachers." This serves as a novel method to protect a model's intellectual property (IP). Experiments demonstrate that while these modified teachers retain their original accuracy, students attempting to distill from them experience a catastrophic performance collapse (over 80% accuracy loss).

**Strengths:**

1. This paper provides a novel identification and formalization of a critical problem (the "distillation trap") that explains why LLM knowledge distillation fails.
2. This paper then proposes an innovative and practical method (Reinforcement Fine-tuning) to control this phenomenon, applying it as a significant new technique for protecting model intellectual property (IP).

**Weaknesses:**

A primary weakness of this paper is the method's foundational trade-off: inducing "distillation traps" inherently introduces nonsensical or hallucinatory artifacts into the teacher model's reasoning process. Although empirical results suggest final answer accuracy is preserved, this degradation of the model's CoT significantly degrades the user experience in any application where reasoning is exposed. More importantly, it erodes trust and harms the model's fundamental interpretability.

Furthermore, the paper's evaluation is narrowly focused on benchmarks characterized by short, definitive answers (e.g., mathematical or factual reasoning). This limitation raises significant doubts about the method's generalizability. For open-ended, generative tasks (such as creative writing or long-article summarization), the distinction between the "reasoning path" and the "final answer" effectively collapses. In such scenarios, it is highly uncertain whether the teacher's final output would remain coherent, or if the hallucinations introduced as "traps" would inevitably corrupt the final output itself.

**Questions:**

N/A

---

### Official Review · Reviewer_X5dZ · 2025-11-02

**Soundness:** 3
**Presentation:** 3
**Contribution:** 3
**Rating:** 2
**Confidence:** 4

**Summary:**

This paper addresses the critical and under-explored phenomenon of Knowledge Distillation (KD) failure in Large Language Models (LLMs). The central contribution is the introduction and rigorous definition of the"Distillation Trap,"which arises when a teacher LLM's high-quality final answer is decoupled from its low-quality, confusing, or misleading intermediate probability distribution. The authors propose a novel, controllable framework using Reinforcement Fine-Tuning with a custom Trap Reward to intentionally engineer "undistillable" teacher models. The RFT objective balances maintaining high task accuracy while maximizing confusion for a student model. Empirical results demonstrate that these engineered teachers can degrade student performance by over 90% across various LLM pairs and datasets. The work offers a profound insight into the mechanics of KD and presents a promising direction for intellectual property protection and model robustness analysis.

**Strengths:**

1.The paper tackles an under-explored yet practically critical issue—the mechanism behind unexpected KD resistance. The concept of controllable undistill-ability is highly original, offering both a new tool for robustness testing and a potent defense for proprietary models.
2.The analysis linking the asymmetry of Forward/Reverse KL Divergence to the distillation trap provides a clear conceptual foundation. This is effectively validated by the proposed RFT-based engineering approach, which successfully dissociates task performance from distill-ability.
3.The experiments are extensive, covering diverse LLM architectures (Teacher-Student pairs) and datasets. The consistency and magnitude of the degradation (up to 90%+ performance drop) are highly convincing and underscore the efficacy of the proposed mechanism.
4.The use of GRPO with LoRA adaptation within the RFT framework is an elegant and computationally feasible design. Algorithm 1 and the implementation details enhance the clarity and potential reproducibility of the work.
5.The findings are directly relevant to next-generation LLM security, providing a fundamental mechanism for model IP protection against unauthorized imitation, thereby opening a vital new research frontier.

**Weaknesses:**

1.Limited Scope of IP Defense and Attacker Model:The claimed IP protection is potentially overstated as the defense assumes a narrow attacker model that relies strictly on soft-label KD (e.g., GKD with JSD). The paper fails to quantify the defense's robustness against more practical attacker strategies, such as:
-Supervised Fine-Tuning (SFT) on Hard Labels:Simply using the teacher's final, correct answer as the ground truth.
-RL Imitation:Directly mimicking the teacher's action policy in an RL setting.
-Top-K Logit Distillation:Filtering the confused logits.
-Recommendation:A quantitative baseline comparing the proposed method against these bypass strategies is essential for a realistic assessment of the defense's efficacy.
2.Insufficient Quantitative Interpretability of the Trap:While qualitative examples of "hallucination-like" outputs are provided, the paper lacks systematic quantitative metrics to characterize the nature of the confusion. Key metrics that should be analyzed include:
-Consistency Measures:e.g., Perplexity gap between the trap output and a clean output.
-Hallucination Rate:Token-level divergence from factual consistency.
-Error Type Analysis:Classification of the confusion (e.g., repetition, semantic drift, logical fallacy).
3.Generaliz ability Across KD Variants and Capacity:The reported results primarily focus on a single KD objective (GKD with JSD). The generality of the "undistillability" across key variants remains unclear:
-KD Objectives:Evaluation must be extended to canonical Forward KL (FKL),Reverse KL (RKL), and Sequence-level KD (SeqKD).
-Student Capacity:The relationship between student size (capacity) and the trap's effectiveness requires further ablation.
4.Sensitivity and Stability Analysis Deficiencies:The critical dependency on hyperparameter selection and the reference model  is not adequately explored:
 Sensitivity:The impact of 's strength (size, pre-training quality, or architectural difference) on the resulting trap reward and strength needs to be rigorously analyzed.
-Training Stability:Crucial factors like the trap weight  and potential reward hacking behaviors (where the teacher exploits the reward signal with nonsensical but high-scoring outputs) require dedicated sensitivity curves and qualitative discussion.
5.Under-explored Ethical and Dual-Use Implications:The Broader Impact section, while present, needs expansion. Intentionally polluting knowledge streams to enforce IP protection introduces significant ethical concerns regarding the injection of unreliability, bias amplification, or deliberate obfuscation into the model's internal representations. This dual-use nature warrants a more comprehensive discussion on responsible deployment.

**Questions:**

1.Please provide quantitative results assessing the effectiveness of the "undistillable" defense when the attacker employs Supervised Fine-Tuning (SFT)on hard labels or RL imitation instead of soft-label KD.
2.Conduct a thorough sensitivity analysis of the trap reward by varying the strength, size, or architectural configuration of the reference model .
3.Does the observed undistillability persist when employing canonical KD objectives such as Forward KL,Reverse KL, or Sequence-level KD?
4.Provide quantitative metrics(e.g., perplexity, n-gram repetition, or a defined hallucination score) to systematically characterize the nature of the confusing outputs generated by the "undistillable" teacher.
5.Detail the training dynamics of the RFT process: specifically, report sensitivity curves for the trap weight  and discuss any observed instances of reward-hacking behaviors.

---

### Note · Authors · 2025-11-12

I have read and agree with the venue's withdrawal policy on behalf of myself and my co-authors.